# The Therapeutic Treatment with the GAG-Binding Chemokine Fragment CXCL9(74–103) Attenuates Neutrophilic Inflammation and Lung Dysfunction during *Klebsiella pneumoniae* Infection in Mice

**DOI:** 10.3390/ijms23116246

**Published:** 2022-06-02

**Authors:** Daiane Boff, Remo Castro Russo, Helena Crijns, Vivian Louise Soares de Oliveira, Matheus Silvério Mattos, Pedro Elias Marques, Gustavo Batista Menezes, Angélica Thomaz Vieira, Mauro Martins Teixeira, Paul Proost, Flávio Almeida Amaral

**Affiliations:** 1Imunofarmacologia, Department of Biochemistry and Immunology, Institute of Biological Sciences, Universidade Federal de Minas Gerais, Belo Horizonte 31270-901, Brazil; daiane.boff@nyulangone.org (D.B.); vivianlouise.soaresdeoliveira@kuleuven.be (V.L.S.d.O.); mmtex.ufmg@gmail.com (M.M.T.); 2Laboratory of Molecular Immunology, Department of Microbiology, Immunology and Transplantation, Rega Institute, KU Leuven, 3000 Leuven, Belgium; helena.crijns@kuleuven.be (H.C.); matheus.mattos@kuleuven.be (M.S.M.); pedro.marques@kuleuven.be (P.E.M.); 3Laboratory of Pulmonary Immunology and Mechanics, Department of Physiology and Biophysics, Institute of Biological Sciences, Universidade Federal de Minas Gerais, Belo Horizonte 31270-901, Brazil; russorc@gmail.com; 4Center of Gastrointestinal Biology, Department of Morphology, Institute of Biological Sciences, Universidade Federal de Minas Gerais, Belo Horizonte 31270-901, Brazil; menezesgb@gmail.com; 5Laboratory of Microbiota and Immunomodulation, Department of Biochemistry and Immunology, Institute of Biological Sciences, Federal University of Minas Gerais, Belo Horizonte 31270-901, Brazil; angelicathomaz@gmail.com

**Keywords:** pneumonia, *Klebsiella pneumoniae*, neutrophils, chemokines, inflammation

## Abstract

*Klebsiella pneumoniae* is an important pathogen associated with hospital-acquired pneumonia (HAP). Bacterial pneumonia is characterized by a harmful inflammatory response with a massive influx of neutrophils, production of cytokines and chemokines, and consequent tissue damage and dysfunction. Targeted therapies to block neutrophil migration to avoid tissue damage while keeping the antimicrobial properties of tissue remains a challenge in the field. Here we tested the effect of the anti-inflammatory properties of the chemokine fragment CXCL9(74–103) in pneumonia induced by *Klebsiella pneumoniae* in mice. Mice were infected by intratracheal injection of *Klebsiella pneumoniae* and 6 h after infection were treated systemically with CXCL9(74–103). The recruitment of leukocytes, levels of cytokines and chemokines, colony-forming units (CFU), and lung function were evaluated. The treatment with CXCL9(74–103) decreased neutrophil migration to the airways and the production of the cytokine interleukin-1β (IL-1β) without affecting bacterial control. In addition, the therapeutic treatment improved lung function in infected mice. Our results indicated that the treatment with CXCL9(74–103) reduced inflammation and improved lung function in *Klebsiella pneumoniae*-induced pneumonia.

## 1. Introduction

Pneumonia is among the leading cause of hospitalization and mortality worldwide [1]. According to the World Health Organization (WHO), it accounts for 15% of all deaths of children under 5 years old [2]. Pneumonia is characterized as an acute infection of the lung parenchyma and can be caused by a wide variety of microorganisms, including bacteria, viruses, and fungi [3]. Among bacteria, *Klebsiella pneumoniae*, an encapsulated, non-motile, lactose-fermenting, anaerobic, and Gram-negative rod, is an important causative agent of hospital-acquired pneumonia (HAP) [4]. Pneumonia associated with *K. pneumoniae* is characterized by a severe and exacerbated acute inflammation of lung tissue with an important recruitment of neutrophils and a high production of cytokines and chemokines [5]. This intense inflammatory response can severely compromise lung function leading to a decrease in compliance and shortness of breath that can progress to respiratory failure and even death [6].

The recruitment of neutrophils to the lungs during infection is essential for bacterial clearance and to avoid its dissemination. Indeed, depletion of neutrophils increases the bacterial burden in the lungs of mice infected with *K. pneumoniae* [7,8]. At the site of infection, activated neutrophils play a fundamental role in bacterial killing by a range of antimicrobial tools [9]. They can directly kill *K. pneumoniae* by releasing granules [10], neutrophil extracellular traps (NETs) [11], antimicrobial peptides [12], and serine proteases [13]. In addition, neutrophils efficiently phagocytose and kill *K. pneumoniae* by producing reactive oxygen species (ROS) [14]. Besides antimicrobial functions, neutrophils also play a role in the immune response against *K. pneumoniae* by the production of cytokines such as tumor necrosis factor (TNF) and the interleukins (IL) IL-1, IL-17, and IL-12 [15,16]. However, neutrophils can act as a double-edged sword because their excessive accumulation and release of those antimicrobial mediators can cause tissue damage and contribute to severe disease development [17,18].

The roles of neutrophils in the progression of pneumonia make them potential targets of neutralization during bacterial infection [19]. The ideal strategy is attenuating their accumulation and destructive potential while maintaining their critical antibacterial properties. Chemokines play an essential role in the migration of neutrophils from blood to the infected lung, and they have emerged as a potential target to inhibit neutrophil accumulation [20]. At this point, CXC chemokines with an ELR (glutamic acid–leucine–arginine) motif (CXCL1, CXCL2, CXCL3, CXCL5, CXCL6, CXCL7, and CXCL8) play a crucial role in neutrophil recruitment by binding to CXC chemokine receptor 1 (CXCR1) and/or CXCR2 [21]. Moreover, chemokines bind to glycosaminoglycans (GAGs) in endothelial cells and generate an immobilized chemokine gradient that directs cell migration [22]. In this context, compounds that compete with chemokines for GAG binding could decrease neutrophil migration to the site of infection [23,24]. Our group previously demonstrated that the fragment of the chemokine CXCL9 consisting of its 30 COOH-terminal amino acids (CXCL9(74–103)) strongly binds to soluble GAGs and GAGs on human microvascular endothelial cells [25,26]. Moreover, this peptide lacks the chemokine domain and, as such, also has no chemotactic activity and does not bind to or signal through the CXCL9 chemokine receptor CXCR3 [27]. By competing with active chemokines for presentation on GAGs, systemic treatment with CXCL9(74–103) successfully inhibits neutrophil migration in murine models of joint inflammation [24,27] and liver damage [28]. Here we demonstrated that the therapeutic treatment with CXCL9(74–103) reduced the recruitment of neutrophils to the lungs without affecting the local bacterial burden. Moreover, treatment resulted in improved lung function in mice infected with *K. pneumoniae*.

## 2. Results

### 2.1. CXCL9(74–103) Binds to the Lung Endothelium In Vivo

CXCL9(74–103) is a fragment of the chemokine CXCL9 consisting of the 30 C-terminal amino acids. We previously demonstrated that this highly positively charged fragment competes with chemokines for GAG binding [24]. To evaluate if CXCL9(74–103) could bind to GAGs on lung endothelium in vivo, we conjugated the peptide site specifically at the NH_2_-terminus with the fluorophore 5(6)carboxytetramethylrhodamine (TAMRA). Then, healthy lung cryosections were immunostained for CD31, an endothelial cell marker, heparan sulfate (HS), and Hoechst, to show nuclei. As depicted in Figure 1A, both the merged image and the Pearson’s coefficient analysis highlighted that CD31 and HS colocalize in the lungs, which does not occur when the same comparison was made between HS and the nuclear Hoechst staining. Next, mice were injected intravenously with the labeled peptide and an anti-CD31 antibody. Thirty minutes after of the injection, the lungs were harvested and immediately imaged under the confocal microscope. We saw extensive colocalization between CD31 and CXCL9(74–103), as visualized in the merged images and confirmed by the elevated Pearson’s coefficient (Figure 1B). In order to confirm that CXCL9(74–103) was also colocalized with CD31 in other organs, we imaged the liver of these mice (Figure 1C). Similar to the staining in the lungs, the peptide co-localizes extensively with CD31 in the liver sinusoids, although some areas display stronger labeling by CXCL9(74–103). In summary, we have provided evidence that CD31-positive endothelial cells, which express HS in the lung, are the same cells that are labeled by CXCL9(74–103) in vivo.

Alternatively, mice injected with anti-CD31 antibody and CXCL9(74–103) labeled with TAMRA were euthanized 6 h or 12 h after the injection to evaluate the stability of peptide binding on the endothelium. The lungs were removed and imaged by intravital microscopy. CXCL9(74–103) was detected in lung tissue only 6 h after the systemic injection, as demonstrated by its colocalization with blood vessels (Figure 2A–C). Since no sign of CXCL9(74–103) was detected 12 h after injection, these data suggest that CXCL9(74–103) can regulate neutrophil diapedesis to the lungs for at least 6 h.

### 2.2. The Treatment with CXCL9(74–103) Decreases Lung Inflammation and Neutrophil Recruitment in Response to Lipopolysaccharide (LPS) In Vivo

The intranasal instillation of LPS triggers an inflammatory response characterized by a huge neutrophil accumulation into the airways [29]. We next evaluated if, by binding to GAGs in the lungs, CXCL9(74–103) could decrease neutrophil migration and inflammation after LPS challenge. Mice were instilled with LPS (25 μg in saline/mouse) and were treated with 100 μL of CXCL9(74–103) 1 mg/mL or phosphate-buffered saline (PBS) as vehicle intravenously 6 h later, with the euthanasia occurring 18 h after the treatment (24 h after LPS challenge) (Figure 3). The negative controls were instilled with saline. The intravenous treatment of control mice instilled with saline with CXCL9(74–103) did not alter leukocyte content in the bronchoalveolar lavage fluid (BALF) (Appendix A). The treatment with CXCL9(74–103) was able to decrease the number of total leukocytes and neutrophils recruited to the airways in response to LPS (Figure 3A,B) but did not affect the recruitment of mononuclear cells (Figure 3C), which was mostly composed of macrophages. In addition, the treatment with CXCL9(74–103) decreased the levels of IL-1β and CXCL2 (Figure 3D,F), but no differences were observed in levels of the chemokines CXCL1 and CXCL6 or total protein concentrations in (BALF) (Figure 3E,G,H). Taken together, these data demonstrated that the therapeutic treatment with CXCL9(74–103) controls lung inflammation by reducing neutrophil accumulation and decreasing IL-1β and CXCL2 production in the tissue.

### 2.3. The Treatment with CXCL9(74–103) Improved Lung Function in LPS-Instilled Mice

The inflammation associated with the intranasal administration of LPS induces alterations in lung capacity and lung mechanics [30]. Since the treatment with CXCL9(74–103) decreased tissue inflammation, we next wondered if the treatment would improve lung function. Expiratory volume at 20 milliseconds (FEV20) and the Tiffeneau–Pinelli index (FEV20/FVC) were analyzed to evaluate airflow limitation. Other parameters evaluated were lung resistance (Rl—measurement of the resistance of the respiratory tract to the airflow movement during normal aspiration and expiration), chord compliance (Cchord—measurement of lung elastic properties), and lung volume, measured by tidal volume (TV—the amount of air ventilated during one respiratory cycle) and minute volume (MV—the amount of air ventilated in one minute). The treatment with CXCL9(74–103) 6 h after the LPS instillation improved the FEV20, Tiffeneau–Pinelli index, lung resistance, and chord compliance when compared to the vehicle-treated group (Figure 4A–C). However, the treatment was not able to significantly improve tidal and minute volumes (Figure 4D–F) when compared to the vehicle-treated group. Altogether, these findings show a direct association between the reduced lung inflammation with improved lung function in LPS-instilled mice treated with CXCL9(74–103). 

### 2.4. The Treatment with CXCL9(74–103) Decreased the Inflammatory Response in Pneumonia Induced by Klebsiella pneumoniae Infection

Since the treatment with CXCL9(74–103) decreased LPS-induced lung inflammation, as represented by reduced neutrophil recruitment and improved lung function, we next tested if the therapeutic treatment with CXCL9(74–103) would reduce lung inflammation and function in a model of pneumonia induced by *K. pneumoniae*. Mice were intratracheally infected with *K. pneumoniae* and the treatment with CXCL9(74–103) or vehicle (saline) occurred intravenously 6 h later (Figure 5). Mice were euthanized 24 h after the infection for the analysis. As shown in LPS-challenged mice, the treatment with CXCL9(74–103) decreased the number of total leukocytes and neutrophils recruited to the airways when compared to vehicle-treated mice (Figure 5A,B). The treatment did not affect the recruitment of mononuclear cells (Figure 5C). In addition, the treatment with CXCL9(74–103) reduced the levels of IL-1β compared to vehicle-treated mice (Figure 5D), while no differences were found in the levels of the chemokines CXCL1, CXCL2, and CXCL6, nor total protein, in BALF fluid (Figure 5E–H). Considering neutrophils are essential cells for bacterial clearance, we wondered if this treatment could affect the bacterial burden in the lungs. To test this hypothesis, we evaluated the bacterial colony-forming units (CFU) in the airways (in BAL fluid) and in the lung tissue. The treatment did not increase bacterial load in both compartments compared to the vehicle-treated group (Figure 5I,J). These results indicate that the therapeutic treatment with CXCL9(74–103) reduced lung inflammation without affecting host bacterial control.

### 2.5. The Treatment with CXCL9(74–103) Improved Lung Function in Pneumonia Induced by Klebsiella pneumoniae

Similar to LPS-induced acute inflammation, pneumonia induced by *K. pneumoniae* also promoted alterations in lung function (Figure 6). It was tested whether the therapeutic treatment with CXCL9(74–103) could improve lung function. Although this treatment did not affect FEV20 or the Tiffeneau–Pinelli index (FEV20/FVC) significantly (Figure 6A–C), treatment with CXCL9(74–103) 6 h after *K. pneumoniae* infection improved chord compliance (Cchord), minute volume (MV), and lung resistance (Rl) when compared with vehicle-treated mice (Figure 6D–F). These results indicated that the therapeutic treatment with CXCL9(74–103) improved lung function after pneumonia induced by *K. pneumoniae* infection.

## 3. Discussion

The treatment of bacterial pneumonia has been a challenge in the last decade due to, among other factors, antimicrobial-resistant strains such as carbapenem-resistant *K. pneumoniae* and the lack of options to prevent the harmful inflammatory response developed in these infections [31,32,33]. We focused our study on the control of the excessive lung inflammation and loss of function typically observed after *K. pneumoniae* infection in mice by targeting neutrophil accumulation in lung compartments. We demonstrated that the therapeutic treatment with a cationic peptide derived from the chemokine CXCL9(74–103) reduced the recruitment of neutrophils in murine models of LPS-induced lung inflammation and *K. pneumoniae*-induced pneumonia. In both models, this treatment also reduced the amount of IL-1β in bronchoalveolar fluid without impairing the clearance of *K. pneumoniae* or altering the number of mononuclear cells in the lungs. Notably, the treatment also improved lung function after these insults. These findings highlight that decreasing neutrophil accumulation in the tissue by controlling the actions of chemokines in the primordial steps for neutrophil diapedesis can prevent excessive tissue inflammation and malfunction.

Neutrophils are essential cells for bacterial elimination during infection. However, their accumulation and secretion of antimicrobial mediators are associated with tissue damage [34,35]. In this context, therapies that target neutrophil migration while maintaining their antimicrobial function are still the main challenge in this field [19,36]. Some strategies to inhibit neutrophil migration in sterile pulmonary diseases have shown to be effective in different experimental models in mice. For example, the antagonism of the chemokine receptor CXCR2 was able to reduce neutrophil recruitment and improved tissue damage in asthma [37], in LPS-induced acute inflammation [38], and in bleomycin-induced lung fibrosis [39]. Because of the promising results in animal models, these CXCR2 antagonists are now being tested in clinical trials for lung diseases [40]. Moreover, an engineered CXCL8 mutant protein (PA401), which exhibited a much higher affinity toward GAGs, showed anti-inflammatory activity in bleomycin-induced lung fibrosis in mice, resulting in a substantial reduction of transmigrated neutrophils in bronchoalveolar lavage [41]. Regarding bacterial pneumonia, the inhibition of neutrophil recruitment has controversial outcomes. Some studies evaluated the prophylactic inhibition of neutrophils as a preventive treatment with G31P, an antagonist of CXCR2, which was shown to decrease neutrophil recruitment without affecting bacterial load in pneumonia induced by *K. pneumoniae* [42]. Moreover, the pre-treatment with vascular adhesion protein-1 (VAP-1/SSAO), an endothelial bound adhesion molecule with amine oxidase activity involved in neutrophil egress from the microvasculature during inflammation, decreased neutrophil migration but increased bacterial load during *K. pneumoniae*-induced pneumonia [43]. In another study, the therapeutic administration of a monoclonal antibody against the granulocyte-colony-stimulating factor receptor (G-CSFR mAb) during *Streptococcus pneumoniae* infection significantly reduced blood and airway neutrophil numbers without affecting bacterial load [44]. In accordance, we demonstrated that the treatment with CXCL9(74–103) decreased neutrophil migration to the lungs without affecting the bacterial clearance by the host after *K. pneumoniae* infection, even when the treatment started 6 h after infection. Our study was limited to the evaluation of the effect of CXCL9 (74–103) at 1 day after the infection. Further studies need to be performed to evaluate later time points. However, we can speculate that the initial recruitment of neutrophils to the lung before the application of CXCL9(74–103) may have an important beneficial effect on the control of *K. pneumoniae*. Therefore, reducing massive neutrophil accumulation by this treatment at later time points may positively preserve lung tissue function without stimulating bacterial growth. In addition, the treatment with CXCL9(74–103) did not impact the number of total macrophages, which are also important cells for the bacterial killing process.

The high production of cytokines and chemokines can be easily detected systemically and locally, and these inflammatory molecules are important biomarkers during the development of acute lung diseases, including the current coronavirus disease 2019 (COVID-19) pandemic [45,46,47]. Pneumonia induced by *K. pneumoniae* or LPS is associated with the overproduction of IL-1β [48,49] and neutrophil-related chemokines such as CXCL1, CXCL2, and CXCL6 [50,51]. Here, mice treated with CXCL9(74–103) presented a substantial reduction of IL-1β concentrations in BALF in pneumonia induced by both *K. pneumoniae* and LPS. The overexpression of IL-1β in lung tissue coincided with increased lung vascular permeability, which was associated with a massive cellular influx and tissue damage [51,52]. Activated macrophages represented the primary source of IL-1β during acute inflammation, and blocking their activation or directly antagonizing IL-1β contributed to reducing tissue inflammation in experimental models of pneumoniae [53,54]. Here, the treatment with CXCL9(74–103) did not reduce macrophage accumulation in BALF, but significantly decreased the number of neutrophils. At this point, some studies demonstrated that neutrophils are necessary for IL-1β production by macrophages [55,56], which may explain the lower detection of this cytokine in CXCL9(74–103)-treated mice. In addition, neutrophils can also be a source of IL-1β during acute inflammation [57]. Thus, in general, these data explain the direct correlation between the reduction of accumulated neutrophils with the low levels of IL-1β after LPS or *K. pneumoniae* stimulation upon CXCL9(74–103) treatment.

During acute inflammation, several chemoattractant molecules are produced locally to guide neutrophil migration to the affected tissue. GAGs have a crucial role in the initial steps of leukocyte migration due to their negatively charged structures, retaining positively charged chemokines on the endothelium. Thus, endothelium-attached chemokines may bind to their chemokine receptors on the rolling cells, modify integrin conformation from an inside to outside manner on leukocytes, and strengthen leukocyte adhesion to endothelial cells [23,58]. At this point, CXCL9(74–103) impairs leukocyte migration by competing with chemokine binding to different types of GAGs [24,27,28,59]. As a consequence, CXCL9(74–103) treatment caused a reduction of neutrophil migration to the tissue after LPS and *K. pneumoniae* insults, reducing lung inflammation as exemplified by the lower detection of IL-1β. However, this treatment did not completely abolish the accumulation of neutrophils in BALF. Thus, the presence of neutrophils, even in lower numbers when compared to non-treated mice, still contributes to the clearance of *K. pneumoniae* once activated by locally produced chemokines. At this point, chemokines not only guide the cells to the site of infection but have an important role in the activation of these cells to deal with infection [60,61]. In a previous study, we showed that CXCL8 stimulation assisted in the clearance of *Staphylococcus aureus* by purified neutrophils [34]. Here, the treatment with CXCL9(74–103) did not reduce the levels of some CXCR2-binding chemokines, which may help to explain why this treatment did not impair the bacterial control.

The lung is the central organ of the respiratory system, and its good functioning is essential to facilitate gas exchange between the environment and bloodstream [51]. Any disturbance of lung homeostasis that occurs in acute inflammation potentially alters its function, which represents an important clinical concern [62,63]. Besides the direct involvement of IL-1β causing lung damage and dysfunction [64], exudative fluids and the excessive accumulation of neutrophils in lung tissue also play critical roles in lung inflammation, damage, and dysfunction during pneumonia [65]. As a result, air space does not inflate properly at higher transpulmonary pressures, reducing the total lung volume and compliance that cause the shortness of *breath and low oxygen supply* [50]. Targeting chemokines and neutrophil migration have been shown as promising strategies to recover lung function after different types of lung insults in sterile [37,66] and viral [67,68] lung inflammation. Here, LPS and *K. pneumoniae* caused profound alterations in respiratory mechanics in mice, and the treatment with CXCL9(74–103) ameliorated some of those parameters, highlighting its beneficial effects to control critical symptoms of acute lung inflammation.

In conclusion, targeting pulmonary GAGs may be an effective strategy to control signs and symptoms of acute lung inflammation, preventing excessive leukocyte accumulation in pulmonary compartments, reducing local production of pro-inflammatory molecules, and ameliorating lung function without affecting bacteria burden.

## 4. Materials and Methods

### 4.1. Mice and Reagents

Eight-to-ten-week-old female C57BL/6J mice were purchased from Centro de Bioterismo da Universidade Federal de Minas Gerais (UFMG). All animals were maintained with filtered water and food ad libitum and kept in a controlled environment. Experiments received prior approval by the animal ethics committee of UFMG (CEUA 347/2019) at 10 February 2020. LPS (Lipopolysaccharide from *Escherichia coli* serotype O:111:B4) were purchased from Sigma-Aldrich (St. Louis, MO, USA). The anti-mouse CD31 APC antibody was purchased from Becton Dickinson (BD, Madrid, Spain). 

### 4.2. Solid-Phase Synthesis of the C-Terminal CXCL9-Derived Peptide

The C-terminal peptide of CXCL9, CXCL9(74–103), was chemically synthesized with fluorenyl methoxycarbonyl (Fmoc) chemistry using an Activo-P11 automated synthesizer (Activotec, Cambridge, UK), as previously described [69]. Part of the material was fluorescently labeled site-specifically at the N-terminus using TAMRA (Merck Millipore, Darmstadt, Germany) [27]. After synthesis, intact synthetic lyophilized peptides were dissolved in 0.1% trifluoroacetic acid (TFA) and purified by RP-HPLC. Peptides were loaded on a 150 × 10 mm Proto 300 C18 column (Higgins Analytical Inc., Mountain View, CA, USA) in 0.1% TFA in water at a flow rate of 4 mL/min and eluted in an acetonitrile gradient in water containing 0.1% TFA. Eluted proteins were detected by splitting 0.7% of the volume of the column effluent to an ion trap mass spectrometer (Amazon SL, Bruker, Bremen, Germany).

### 4.3. Intravital Microscopy

For intravital imaging, mice were injected i.v. with 100 μL of TAMRA-labeled CXCL9(74–103) at 1 mg/mL, and 10 min prior to euthanasia were injected with anti-CD31 APC or anti-CD31 Alexa Fluor 488. After 30 min, 6 h or 12 h of peptide injection, the lungs were removed and imaged using a 25× objective on a Dragonfly spinning-disk confocal microscope (Andor Technology, Concord, MA, USA) or on a Nikon Eclipse Ti microscope. An automated device controlled the z-position and 10× objectives were used on the required resolution. The analysis was performed using Volocity 6.3 software (PerkinElmer, Waltham, MA, USA).

### 4.4. Bacterial Strain

The bacterium *Klebsiella pneumoniae* ATCC 27736 has been kept at the Department of Microbiology (UFMG), and its pathogenicity was stimulated by 10 passages in C57BL/6 mice. Bacteria were frozen in the logarithmic growth phase and kept at −80 °C at a 1 × 10^9^ CFU/mL concentration in tryptic soy broth (Difco, Detroit, MI, USA) containing 10% (vol/vol) glycerol until use.

### 4.5. LPS-Induced Acute Lung Inflammation

As previously described [70], mice were anesthetized with ketamine–xylazine (50 mg/mL and 0,02 mg/mL; intra-peritoneally) and 30 μL of saline or LPS (25 μg per mouse) was instilled intranasally. Groups of mice were treated with 100 μL of CXCL9(74–103) 1 mg/mL or saline i.v. 6 h after the instillation. At 24 h after the LPS instillation, mice were euthanized by anesthetic overdose.

### 4.6. Klebsiella pneumoniae Lung Infection

The lung infection was performed as previously described [71]. Briefly, the bacteria were cultured for 20 h at 37 °C prior to inoculation. The concentration of bacteria in broth was routinely determined by serial 1:10 dilutions. A total of 100 μL of each dilution was plated onto MacConkey agar (Difco, Detroit, MI, USA) and incubated for 24 h at 37 °C before colonies were counted. Each animal was anesthetized intra-peritoneally with 0.2 mL of a solution containing xylazine (0.02 mg mL), ketamine (50 mg mL), and saline in the relative composition of 1:0.5:3. The trachea was exposed and 25 μL of a suspension containing 1 × 10^6^ CFU of *K. pneumoniae* or saline was administered with a 26-gauge needle. The skin was closed with surgical staples. Groups of mice were treated with 100 μL of CXCL9(74–103) 1 mg/mL or saline i.v. 6 or 12 h after the infection. At 24 h after the challenge, mice were euthanized by anesthetic overdose.

### 4.7. Bronchoalveolar Lavage Fluid (BALF) and CFU Counts

Mice were euthanized with a lethal solution of ketamine/xylazine (180 and 12 mg/kg, respectively), and BALF was collected by inserting and collecting three times 0.5 mL aliquots of PBS, through a 1.7 mm catheter in a 1 mL syringe. After centrifugation, cell pellets were used for total and differential leukocyte counts. The number of total leukocytes was determined by counting them in a modified Neubauer chamber after staining with Turk’s solution. Differential counts were obtained from cytospin preparations (Shandon III) by evaluating the percentage of each leukocyte on a slide stained with May–Grünwald–Giemsa stain and examined by light microscopy. BALF supernatants were used for cytokine and chemokine measurements. The bacterial load in the lungs and BALF was accessed by serial dilutions (1:10) in sterile saline, and 10 μL of each dilution was plated onto MacConkey agar and incubated for 24 h at 37 °C to determine the number of CFU.

### 4.8. Measurement of Chemokines and Cytokines

The cytokine IL-1β (cat. no. DY401-5) and the chemokines CXCL1 (cat. no. DY453-05), CXCL2 (cat. no. DY452-05) and CXCL6 (cat. no. MX000) were measured from the BALF supernatants by ELISA following the manufacturer’s instructions (R&D Systems, Minneapolis, MN, USA).

### 4.9. Assessment of Respiratory Mechanic Dysfunction

As previously described, mice were tracheostomized, placed in a body plethysmograph, and connected to a computer-controlled ventilator (Forced Pulmonary Maneuver System; Buxco Research Systems, Wilmington, NC, USA) [72]. Under mechanical respiration, the tidal volume (TV), volume per minute (MV), and lung resistance (Rl) were determined by the resistance and compliance test. To measure the chord compliance (Cchord), lungs were inflated to a standard pressure of +30 cm H_2_O and then slowly exhaled until a negative pressure of −30 cm H_2_O is reached, and Cchord was evaluated at +10 cm H_2_O. The fast-flow volume maneuver was performed, and lungs were first inflated to +30 cm H_2_O and immediately afterward submitted to a highly negative pressure to enforce expiration until −30 cm H_2_O. The forced vital capacity (FVC) and forced expiratory volume at 20 ms (FEV20) were recorded during this maneuver, and the Tiffeneau–Pinelli index (FEV20/FVC) was calculated using these two variables. Suboptimal maneuvers were rejected, and for each test in every single mouse, at least three acceptable maneuvers were conducted to obtain a reliable mean for all numeric parameters.

### 4.10. Cryosections and Immunostainings

Lungs were harvested, embedded in optimal cutting temperature media, and snap-frozen in liquid nitrogen. Then, 10 μm thickness sections were made using a cryostat (Cryo-Star HM 560, Thermo Scientific, Walldorf, Germany). The sections were fixed with paraformaldehyde (PFA) 4%, and unspecific labeling was blocked with HBSS supplemented with 1% Fc block and 10% fetal calf serum. Primary anti-CD31 (10 μg/mL, R&D Systems, Minneapolis, MN, USA) and anti-HS (10 μg/mL, Amsbio, Cambridge, MA, USA; clone F58-10E4) antibodies were added overnight at 4 °C. Subsequently, the sections were washed with HBSS, and the secondary antibodies were added for 2 h. Images were taken using a 25× objective in a Dragonfly spinning-disk confocal microscope (Andor Technology, Concord, MA, USA). The pictures were analyzed using FIJI software.

### 4.11. Statistical Analyses

Data are expressed as medians and analysis were performed using the statistical software GraphPad Prism 8.0 (GraphPad Software, San Diego, CA, USA). Differences between means were evaluated using the ANOVA test, followed by the Newman–Keuls post-test. Results with *p* < 0.05 were considered significant.

## Figures and Tables

**Figure 1 ijms-23-06246-f001:**
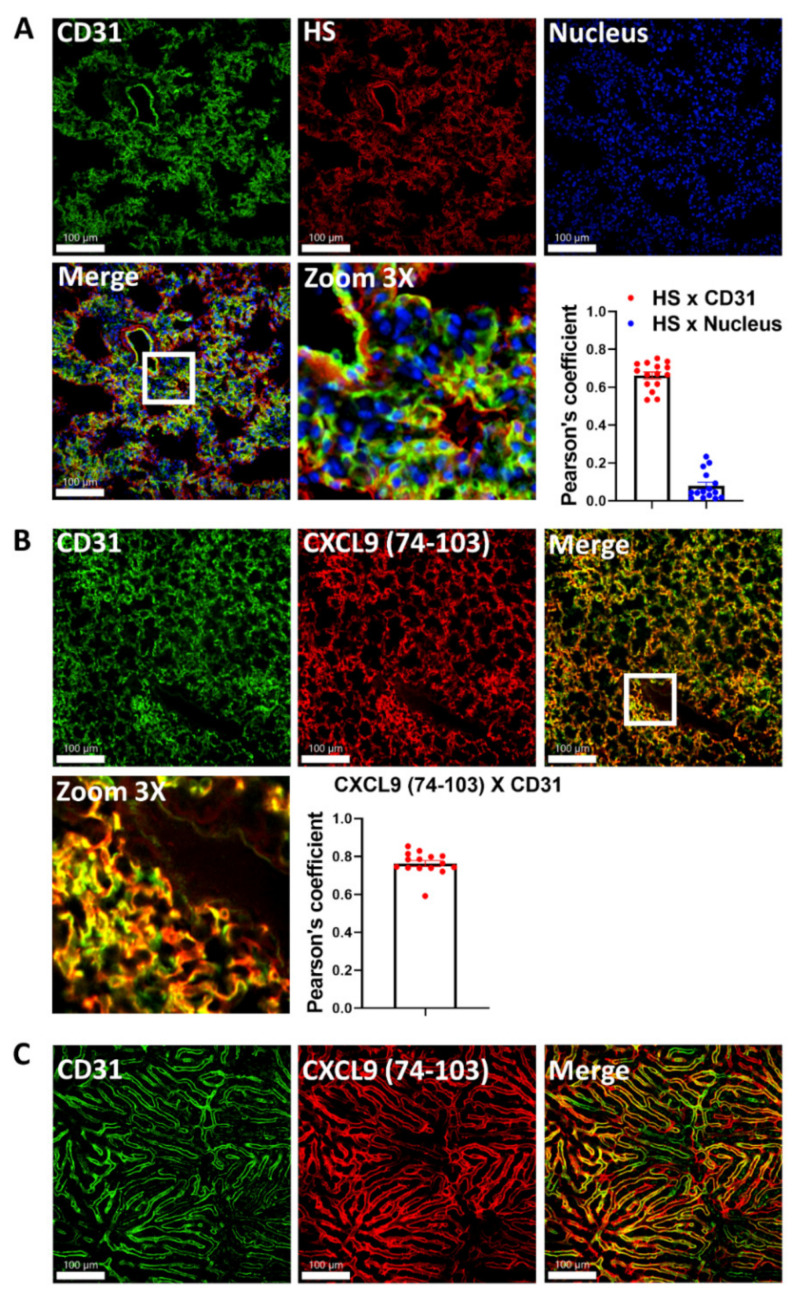
Both CXCL9 (74–103) and heparan sulfate are colocalized with endothelial cells. (**A**) healthy lung cryosections were immunostained for CD31 (green), HS (red), and the nucleus (blue). (**B**) Mice were intravenously injected with TAMRA-labeled CXCL9(74–103) and Alexa Fluor 488-conjugated anti-CD31 (200 μg/kg) and, after 30 min, lungs were harvested and imaged by confocal microscopy. (**C**) The liver of the same mouse used in panel B was imaged under the confocal microscope highlighting the colocalization between CXCL9(74–103) and CD31 also in the liver. The white squares show the area that was zoomed. Red points = HS x CD31; blue points = HS x nucleus; Scale bars = 100 μm.

**Figure 2 ijms-23-06246-f002:**
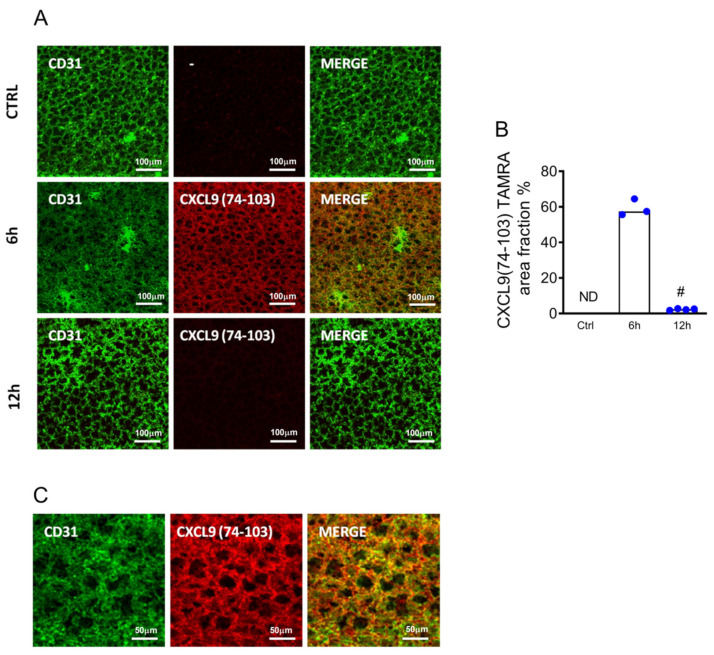
Binding of CXCL9(74–103) to lung endothelium. Mice were injected intravenously with TAMRA-labeled CXCL9(74–103), and lung tissues were collected after 6 h and 12 h for confocal analysis. About 5 min before euthanasia, mice were injected intravenously with allophycocyanin (APC)-labeled anti-CD31 antibodies. (**A**) Lungs were removed and imaged to evaluate the binding of CXCL9(74–103) in the lung endothelium, and (**B**) the percentage of TAMRA area fraction was calculated (blue points). (**C**) A higher magnification of colocalization of CD31 and CXCL9(74–103) 6 h after the injection. Data are shown as median from one representative out of two independent experiments. # *p* < 0.05 when compared with the 6 h; *n* = 3–4 mice per group.

**Figure 3 ijms-23-06246-f003:**
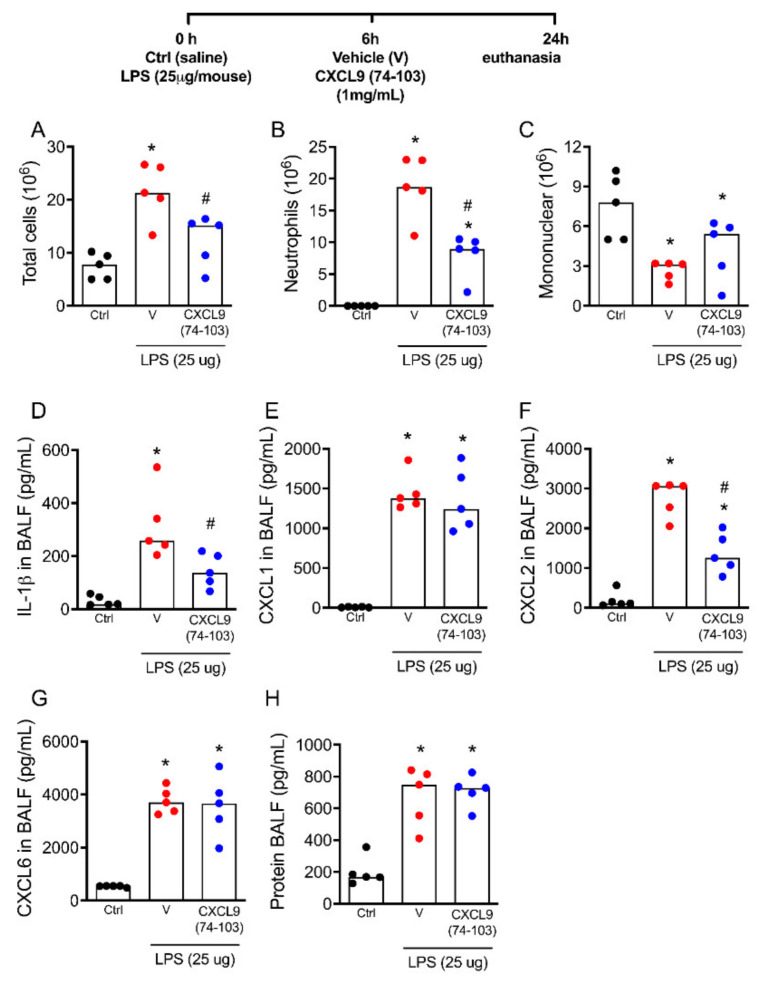
The treatment with CXCL9(74–103) decreases lung inflammation after LPS instillation. Mice were instilled with LPS (25 ug/mouse) or saline (Ctrl), and after 6 h, mice were treated intravenously with 100 μL of CXCL9(74–103) 1 mg/mL or vehicle (PBS). After 18 h, mice were euthanized, and the number of (**A**) total leukocytes, (**B**) neutrophils, and (**C**) mononuclear cells was evaluated in bronchoalveolar lavage fluid (BALF). (**D**) Levels of IL-1β and the chemokines (**E**) CXCL1, (**F**) CXCL2, (**G**) CXCL6, and (**H**) total protein concentrations were also evaluated in the BALF. Black points = control; red points = vehicle; blue points = CXCL9(74–103). Data are shown as the median from one representative out of three independent experiments. * *p* < 0.05 when compared to control group; # *p* < 0.05 when compared with the vehicle group; *n* = 5 mice per group.

**Figure 4 ijms-23-06246-f004:**
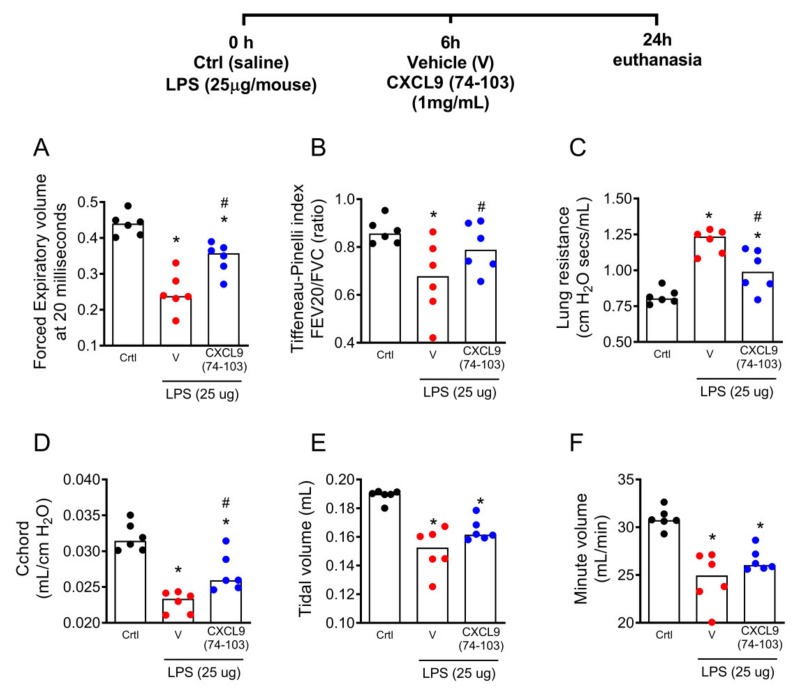
The treatment with CXCL9(74–103) improved lung function after LPS instillation. Mice were instilled with LPS (25 ug/mouse), and after 6 h, mice were treated intravenously with 100 μL of CXCL9(74–103) at 1 mg/mL or vehicle (PBS). Lung function was evaluated after 24 h of LPS challenge. (**A**) Expiratory volume at 20 milliseconds (FEV20), (**B**) Tiffeneau–Pinelli index (FEV20/FVC), (**C**) resistance (Rl), (**D**) chord compliance (Cchord), (**E**) tidal volume (TV), and (**F**) minute volume (MV). Black points = control; red points = Vehicle; blue points = CXCL9(74–103). Data are shown as the median from one representative out of three independent experiments. * *p* < 0.05 when compared to control group; # *p* < 0.05 when compared with the vehicle group; *n* = 5 mice per group.

**Figure 5 ijms-23-06246-f005:**
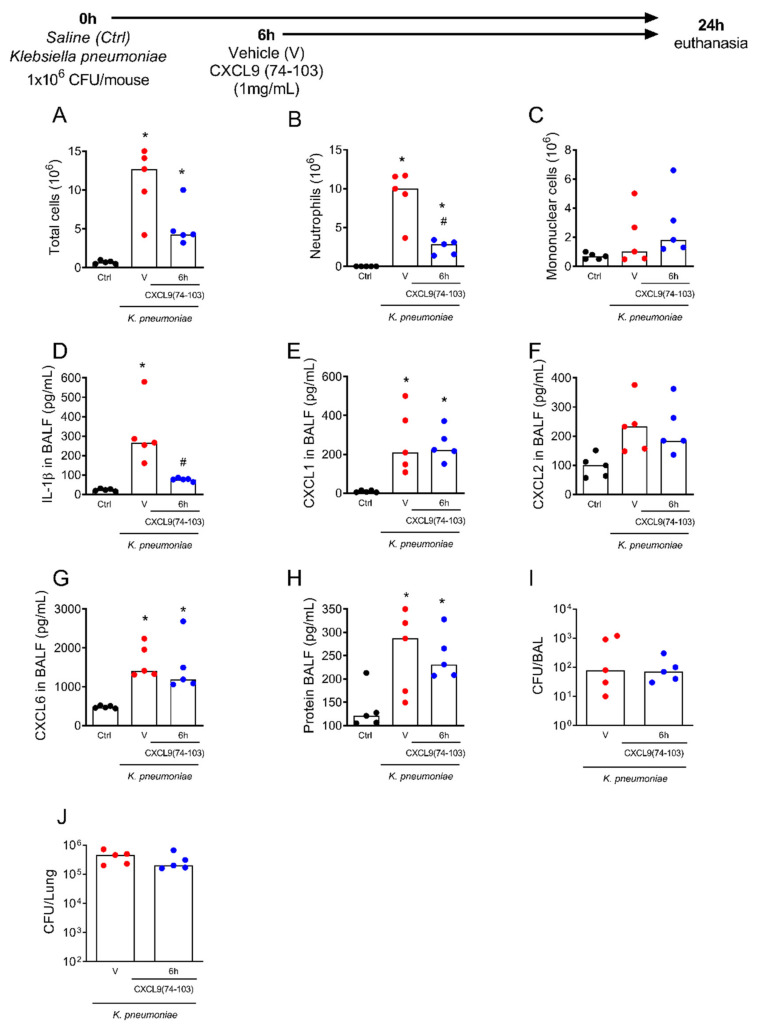
The treatment with CXCL9(74–103) decreased inflammation after pneumonia induced by *Klebsiella pneumoniae*. Mice were infected with *K. pneumoniae* (1 × 10^6^ CFU/mouse) by intratracheal injection, and after 6 h, mice were treated intravenously with 100 μL of CXCL9(74–103) at 1 mg/mL or vehicle (PBS). Control mice received a saline injection into the trachea. Mice were euthanized 24 h after intratracheal challenge, and (**A**) the number of total leukocytes, (**B**) neutrophils, and (**C**) mononuclear cells were evaluated in BALF. (**D**) Levels of IL-1β, and of the chemokines (**E**) CXCL1, (**F**) CXCL2, and (**G**) CXCL6, as well as (**H**) total protein were measured, in the BALF. (**I**) CFU counts in BALF and (**J**) CFU counts in the lung tissue. Black points = control; red points = vehicle; blue points = 6 h CXCL9(74–103). Data are shown as the median from one representative out of three independent experiments. * *p* < 0.05 when compared to control (Ctrl) group; # *p* < 0.05 when compared with the vehicle (v) group; *n* = 5 mice per group. BALF—Bronchoalveolar fluid.

**Figure 6 ijms-23-06246-f006:**
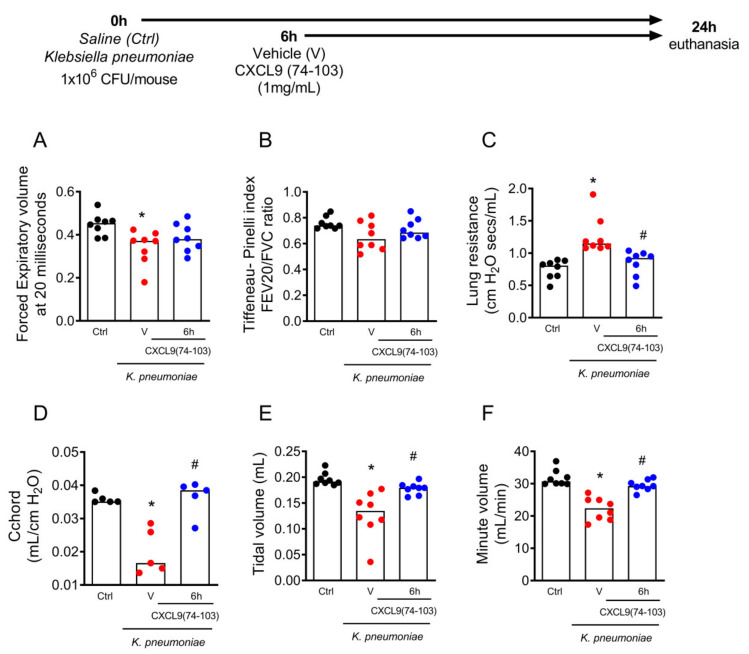
The treatment with CXCL9(74–103) improved lung function after pneumonia induced by *Klebsiella pneumoniae*. Mice were infected with *Klebsiella pneumoniae* (1 × 10^6^ CFU/mouse) by intratracheal injection and after 6 h, mice were treated intravenously with 100 μL of CXCL9(74–103) at 1 mg/mL or vehicle (PBS). Lung function was evaluated 24 h after the challenge. (**A**) Expiratory volume at 20 milliseconds (FEV20), (**B**) Tiffeneau–Pinelli index (FEV20/FVC), (**C**) resistance (Rl), (**D**) chord compliance (Cchord), (**E**) tidal volume (TV), and (**F**) minute volume (MV). Black points = control; red points = vehicle; blue points = 6 h CXCL9(74–103). Data are shown as the median from one representative out of three independent experiments. * *p* < 0.05 when compared to the control (Ctrl) group; # *p* < 0.05 when compared with the vehicle (v) group; *n* = 5–8 mice per group.

## Data Availability

All data generated in this study are included in the manuscript.

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
