# Peer review of "The Therapeutic Treatment with the GAG-Binding Chemokine Fragment CXCL9(74–103) Attenuates Neutrophilic Inflammation and Lung Dysfunction during Klebsiella pneumoniae Infection in Mice"

_ijms, 2022, doi:10.3390/ijms23116246_

Round 1
Reviewer 1 Report
The manuscript "The therapeutic treatment with the chemokine fragment CXCL9 (74-103) decreases inflammation and improves lung function in pneumonia induced by Klebsiella pneumoniae " aimed to test the effect of the anti-inflammatory properties of the CXCL9 (74-103) in Klebsiella pneumoniae/ LPS induced pneumonia like disease characteristics in C57BL6 WT mice. These data demonstrated that as compared to
vehicle, CXCL9 (74-103) treatment caused marked reduction in Klebsiella pneumoniae /LPS- mediated increased airway recruitment of PMNs, IL-1β production, and improvement in the lung function. However, such approaches have not shown any impacton bacterial control. The manuscript is concisely written but poorly designed. The conclusions drawn based on presented data are confusing to large extent.
Here are several of the major criticism, which Authors should use to improve the fluence of the text with a revision of the paragraphs, figures by providing the missing data and their right explanation in the discussion.
1. The data shown in Figures 2, 3, 4, and 5 describe the results of condition (1) C57BL6 mice +LPS → → → + CXCL9(74-103) 100 μg/100 μL or vehicle (PBS) but missing the proper results of condition (2) C57BL6 mice+ Saline→ → → + CXCL9(74-103) 100 μg/100 μL or vehicle (PBS). Authors must include these missing data. They may keep all the condition 1 data in main figures and condition 2 data in Supplementary figures.
2. The authors mentioned in the results " To evaluate if CXCL9(74-103) could bind to GAGs on lung endothelium in vivo, we conjugated the peptide site-specifically at the NH2-terminus with the fluorophore 5(6) carboxytetramethylrhodamine (TAMRA). Mice were injected intravenously with the labeled peptide and an anti-CD31 antibody as a marker for blood vessels. Mice were euthanized 6 or 12 hours
after the injection. The lungs were removed and imaged by intravital microscopy.
CXCL9(74-103) was detected in lung tissue only 6 hours after the systemic injection, as demonstrated by its colocalization with blood vessels (Figure 1A-C). To justify the binding of CXCL9 (74-103) to GAGs on lung endothelium, Authors need to add the lung tissue colocalization data for GAGs, (e,g., dermatan sulfate/ heparan sulfate), TAMARA, and/ or CD31.
3. In results, Authors have again mentioned that We next evaluated if, by binding to GAGs in the lungs, CXCL9(74-103) could decrease neutrophil migration and inflammation after LPS challenge. But the data shown in figure 2 not have any direct involvement of GAGSs. Authors need to explain it and provide the supporting data.
4. Authors have shown CXCL9 (74-103) treatment caused the reduction in Klebsiella pneumoniae/ LPS induced BAL increases of total cells and the PMNs (Figure 2 and 4). CXCL9 is potential chemoattractant for several of the other immune cells, (e.g., T cells, macrophages, and dendritic cells; PMID: 25432064, PMID: 34884512, PMID: 24079945, and PMID: 31799781). However, the cell count shown in Figure 2 and 4 have not justified by providing the PMN images. Authors should provide the supported data (e.g., microscopic images and/or FACS data) confirming the cell count alteration happens only in PMN population and the related interpretation in the discussion.
Reviewer 2 Report
Authors investigated the effects of treatment with CXCL9 fragment in a. mice model of Klebsiella pneumonia. They found that treatment decreased neutrophil migration to the airways and the production of the cytokine IL-1β without affecting bacterial control, together with improved lung function in infected mice. The manuscript is interesting, but some points need to be addressed.
-One limitation of the study is the limited (24 hours after challenge) time of observation, so that it is not possible to predict the clinical meaning of the observation.
-Authors speculate that the initial recruitment of neutrophils to the lung before application of CXCL9(74-103) may have an important beneficial effect on the control of K. pneumoniae. They should discuss that further studies should explore the right timing and schedule of CXCL9(74-103) treatment, in order to control lung inflammation and bacterial growth (24 hours observation is too short time)
-In the LPS model, the treatment improved airway obstruction, but it was not able to significantly improve the dynamic compliance, when compared to vehicle-treated group. The reverse was observed after Klebsiella infection. These findings mean that the treatment improved airway obstruction in the LPS model and the functional consequences on small airways of lung parenchyma inflammation following Klebsiella infection. Authors should have measured static compliance, as this functional parameter better depends on the effects of pneumonia on the functional properties of lung (i.e. Loss of surfactant asin ARDS, decreased lung elasticity, Pulmonary oedema, Alveolar de-recruitment etc).
Author Response
Reply to Reviewer 2
Comments and Suggestions for Authors
Authors investigated the effects of treatment with CXCL9 fragment in a. mice model of Klebsiella pneumonia. They found that treatment decreased neutrophil migration to the airways and the production of the cytokine IL-1β without affecting bacterial control, together with improved lung function in infected mice. The manuscript is interesting, but some points need to be addressed.
1-One limitation of the study is the limited (24 hours after challenge) time of observation, so that it is not possible to predict the clinical meaning of the observation.
Reply: We thank the reviewer for the comment. Our group previously demonstrated (PMID: 29515566) that the peak of neutrophil recruitment to the lungs in this model is 24 hours after bacterial infection with a significant decrease at 48 hours. Accordingly, in order to evaluate the effect of CXCL9 (74-103) treatment in the neutrophil recruitment we performed the experiments 24 hours after the bacterial challenge. However, we agree that new experiments should be performed in the future to evaluate the effect of the treatment at different time points. A statement was added to the discussion section addressing the need for future experiments (on page 13).
-Authors speculate that the initial recruitment of neutrophils to the lung before application of CXCL9(74-103) may have an important beneficial effect on the control of K. pneumoniae. They should discuss that further studies should explore the right timing and schedule of CXCL9(74-103) treatment, in order to control lung inflammation and bacterial growth (24 hours observation is too short time)
Reply: We thank the reviewer for the comment. As indicated in our answer to comment 1 a statement was added to the discussion section addressing the need for future experiments to evaluate the effect of the treatment at later time points.
-In the LPS model, the treatment improved airway obstruction, but it was not able to significantly improve the dynamic compliance, when compared to vehicle-treated group. The reverse was observed after Klebsiella infection. These findings mean that the treatment improved airway obstruction in the LPS model and the functional consequences on small airways of lung parenchyma inflammation following Klebsiella infection. Authors should have measured static compliance, as this functional parameter better depends on the effects of pneumonia on the functional properties of lung (i.e. Loss of surfactant asin ARDS, decreased lung elasticity, Pulmonary oedema, Alveolar de-recruitment etc).
Reply: We are grateful to reviewer-2 for raising this valuable suggestion. Accordingly, we revisited the data obtained by the spirometry system in order to assess static compliance. The data revealed that static compliance drops in both models in vehicle-treated mice groups, however, the treatment with CXCL9(74-103) was able to partially reverse the loss of static compliance in the LPS model, and fully reverse this loss of compliance in the K. pneumoniae model. Furthermore, these differences between loss of compliance in vehicle-treated mice were observed in both models, reflecting the magnitude of neutrophilic inflammation in each of them. Thus, LPS instillation causes an inflammatory response with a twice higher neutrophilic influx into the BAL (20 x 106 neutrophils) when compared to neutrophilic inflammation during K. pneumoniae infection (12 x 106 neutrophils), and the compliance loss was more significant in LPS then K. pneumoniae infection. In conclusion, looking at this time window of 24h post LPS or K. pneumoniae infection, both lead to loss of compliance, and the intensity of this depends on how substantial the neutrophil influx is in these different inflammatory models of the airway disease. We agree with this reviewer, and in the manuscript we replaced the dynamic compliance graphs of both models with static compliance, measured at +10 cm H2O.
Reviewer 3 Report
The manuscript ijms-1707341 devoted the actual field of the molecular biology, namely the possible therapeutic treatment of pneumonia with the chemokine fragment CXCL9(74-103) and can be interested to the specialists working in this field. The opinion of authors is clear and based on a wide range of publications. I am personally impressed by the structure of the article, the systematization of scientific data and the sequence of its presentation. The paper fit the Journal scope and formal requirements.
My decision is accept.
Author Response
We thank the reviewer for accepting our paper for publication.
Round 2
Reviewer 1 Report
Dear Authors,
You have done good job posting right clarification and new data in the revised MS. I am now intensely recommending the publication of your article {The therapeutic treatment with the chemokine fragment CXCL9 (74-103) decreases inflammation and improves lung function in pneumonia induced by Klebsiella pneumoniae}.
Best Luck,
This manuscript is a resubmission of an earlier submission. The following is a list of the peer review reports and author responses from that submission.
Round 1
Reviewer 1 Report
The manuscript is very interesting for a general audience being on an important public health topic.
General comment.
The abreviatures should be written in full the first time they appear in the text.
The manuscript should be revised for typo errors, and font format.
Specific minor comments
Lines 307-309. In addition, given therapeutically, CXCL9(74-103) did not abolish neutrophil accumulation in lung tissue, which helps to explain why there was no loss in bacterial control in the case of K. pneumoniae infection. The sentence is confusing consider revising to improve clarity. This is very important since is the final conclusion.
Line 324-25. After synthesis, intact synthetic peptides were purified by RP-HPLC Describe the procedure, colum, mobile phase, volume injected, detector, etc.
Line 344. i.p. Probably is intra-peritoneal but it should be clear what it is.
Line 345. 100 μg/100 μL this is a odd concentration unit consider using μg/mL
Line 346. Consider replacing killed by sacrificed or euthanized
Line 372. At least give the reference of the kit used so that others could reproduce the experiment
Reviewer 2 Report
The manuscript "The therapeutic treatment with the chemokine fragment CXCL9 (74-103) decreases inflammation and improves lung function in pneumonia induced by Klebsiella pneumoniae " aimed to test the effect of the anti-inflammatory properties of the CXCL9 (74-103) in Klebsiella pneumoniae/ LPS induced pneumonia like disease characteristics in C57BL6 WT mice. These data demonstrated that as compared to vehicle, CXCL9 (74-103) treatment caused marked reduction in Klebsiella pneumoniae /LPS- mediated increased airway recruitment of PMNs, IL-1β production, and improvement in the lung function. However, such approaches have not shown any impact on bacterial control. The manuscript is concisely written but poorly designed. The conclusions drawn based on presented data are confusing to large extent.
Here are several of the major criticism, which Authors should use to improve the fluence of the text with a revision of the paragraphs, figures by providing the missing data and their right explanation in the discussion.
- The data shown in Figures 2, 3, 4, and 5 describe the results of condition (1) C57BL6 mice +LPS → → → + CXCL9(74-103) 100 μg/100 μL or vehicle (PBS) but missing the proper results of condition (2) C57BL6 mice+ Saline→ → → + CXCL9(74-103) 100 μg/100 μL or vehicle (PBS). Authors must include these missing data. They may keep all the condition 1 data in main figures and condition 2 data in Supplementary figures.
- The authors mentioned in the results " To evaluate if CXCL9(74-103) could bind to GAGs on lung endothelium in vivo, we conjugated the peptide site-specifically at the NH2-terminus with the fluorophore 5(6) carboxytetramethylrhodamine (TAMRA). Mice were injected intravenously with the labeled peptide and an anti-CD31 antibody as a marker for blood vessels. Mice were euthanized 6 or 12 hours after the injection. The lungs were removed and imaged by intravital microscopy. CXCL9(74-103) was detected in lung tissue only 6 hours after the systemic injection, as demonstrated by its colocalization with blood vessels (Figure 1A-C). To justify the binding of CXCL9 (74-103) to GAGs on lung endothelium, Authors need to add the lung tissue colocalization data for GAGs, (e,g., dermatan sulfate/ heparan sulfate), TAMARA, and/ or CD31.
- In results, Authors have again mentioned that We next evaluated if, by binding to GAGs in the lungs, CXCL9(74-103) could decrease neutrophil migration and inflammation after LPS challenge. But the data shown in figure 2 not have any direct involvement of GAGSs. Authors need to explain it and provide the supporting data.
- Authors have shown CXCL9 (74-103) treatment caused the reduction in Klebsiella pneumoniae/ LPS induced BAL increases of total cells and the PMNs (Figure 2 and 4). CXCL9 is potential chemoattractant for several of the other immune cells, (e.g., T cells, macrophages, and dendritic cells; PMID: 25432064, PMID: 34884512, PMID: 24079945, and PMID: 31799781). However, the cell count shown in Figure 2 and 4 have not justified by providing the PMN images. Authors should provide the supported data (e.g., microscopic images and/or FACS data) confirming the cell count alteration happens only in PMN population and the related interpretation in the discussion.
